# Combined Selenium and Zinc Biofortification of Bread-Making Wheat under Mediterranean Conditions

**DOI:** 10.3390/plants10061209

**Published:** 2021-06-14

**Authors:** Dolores Reynolds-Marzal, Angelica Rivera-Martin, Oscar Santamaria, Maria J. Poblaciones

**Affiliations:** 1Department of Agronomy and Forest Environment Engineering, University of Extremadura, Avenida Adolfo Suárez s/n, 06007 Badajoz, Spain; lolreymar@gmail.com (D.R.-M.); angelicarm@unex.es (A.R.-M.); 2Department of Construction and Agronomy, University of Salamanca, Avenida Cardenal Cisneros 34, 49029 Zamora, Spain; osantama@usal.es

**Keywords:** sodium selenate, zinc sulfate, cereal, rainfed conditions, forage yield

## Abstract

Millions of people worldwide have an inadequate intake of selenium (Se) and zinc (Zn), and agronomic biofortification may minimise these problems. To evaluate the efficacy of combined foliar Se and Zn fertilisation in bread making wheat (*Triticum aestivum* L.), a two-year field experiment was established in southern Spain under semi-arid Mediterranean conditions, by following a split-split-plot design. The study year (2017/2018, 2018/2019) was considered as the main-plot factor, soil Zn application (50 kg Zn ha^−1^, nor Zn) as a subplot factor and foliar application (nor Se, 10 g Se ha^−1^, 8 kg Zn ha^−1^, 10 g Se ha^−1^ + 8 kg Zn ha^−1^) as a sub-subplot factor. The best treatment to increase both Zn and Se concentration in both straw, 12.3- and 2.7-fold respectively, and grain, 1.3- and 4.3-fold respectively, was the combined foliar application of Zn and Se. This combined Zn and Se application also increased on average the yield of grain, main product of this crop, by almost 7%. Therefore, bread-making wheat seems to be a very suitable crop to be used in biofortification programs with Zn and Se to alleviate their deficiency in both, people when using its grain and livestock when using its straw.

## 1. Introduction

Cereals are the most important crops for both animal feed and human nutrition, supplying between 25% and 90% of their daily energy needs. Among these, wheat is one of the most important, being grown in 120 countries, China, India and Russia as the main producers, with a harvested area of around 220 million ha and a production of more than 770 million Mg [1]. Its relevance lies in the fact that around 82% of wheat grain is made up of carbohydrates, with more than 60% being starch, with an adequate protein content [2]. However, the vitamin and mineral content is generally low [3]. This low mineral content is aggravated by the relatively high content of the anti-nutrient phytate that wheat has, which hinders the absorption of nutrients such as Ca, Fe, Mg, Se and Zn [4,5].

Nowadays, mineral malnutrition, or hidden hunger, is a global problem affecting around 60% of the world’s population, with Fe, I, Se and Zn deficiencies being the most pronounced [6]. The main cause of these deficiencies is the low bioavailability of these nutrients into soil, which produce crops with an inadequate amount of these nutrients in their edible parts. Soils in the semi-arid Mediterranean area have generally low concentrations of both Se and Zn, especially in those of the Southwest of the Iberian Peninsula, which are classified according to [7,8] as deficient to marginal in available Se (<27 µg Se kg^−1^) [9,10,11] and deficient in available Zn (<0.5 mg Zn-DTPA kg^−1^) [12]. 

Selenium is not considered an essential nutrient for angiosperm plants, but it is for animals and humans, where it is a key component of more than 30 selenoproteins or selenoenzymes [13]. It is involved in cell protective processes and is related to the proper functioning of the immune and endocrine systems [14,15]. Its deficiency is linked to oxidative stress, epilepsy, asthma, reduced male fertility, depression of the immune system and the increased risk of certain cancers, such as rectal, liver, prostate and colon cancer [13,16]. On the other hand, Zn is one of the most important trace elements for all living organisms, including plants, in which it plays a particularly important role during periods of rapid growth [17,18]. In animals and humans, it is present in high concentrations in all body tissues, and is involved in many vital functions [19,20]. Its deficiency is associated with diseases such as anemia, anorexia, cancer, gastrointestinal and kidney problems, immune system dysfunction, delayed bone and sexual maturation and DNA damage, as well as being linked to certain types of cancer [21,22]. Due to the antiviral and immune-boosting properties of Se and Zn, recent studies have linked Se levels to the severity of the infection of SARS-CoV-2 [23], and have proposed Se [24] and Zn supplementation [25,26] as treatments to alleviate their symptoms.

One of the most effective remediation strategies to alleviate this problem is agronomic biofortification, i.e., increasing the bioavailable concentration of nutrients in the edible parts of plants through agronomic intervention [27]. While for Se there is some consensus that foliar application of 10 g Se ha^−1^ applied as sodium selenate at anthesis is the most efficient in semi-arid conditions [9,10,28,29], for Zn, there is more controversy. In general, it is considered that zinc sulphate (ZnSO_4_-7H_2_O) is the most widely used fertiliser. However, while the Zn soil incorporation before sowing at a rate of 50 kg ha^−1^ increases mainly grain productivity, the foliar application of 4–8 kg ha^−1^ at the start of flowering seems to be more efficient in increasing Zn concentrations in grain. Therefore, the combined soil and foliar application is considered as a suitable option by [12,30].

However, the information regarding the combined biofortification of Se and Zn, which may allow to alleviate their intake deficiency simultaneously and might reduce application costs for farmers, is very limited, and mainly based on trials carried out under greenhouse conditions [31]. Under in-field conditions, such combined application of Zn and Se under semiarid Mediterranean climate has already demonstrated a high accumulation of those micronutrients in forage peas [32]. However, the effect of the combined Zn and Se biofortification on bread wheat, a crop of global importance, remains unknown under these semiarid conditions, where the irregularity of rainfall could substantively influence its efficiency. The general aim of this study is to contribute to achieve a basic crop in human nutrition with a high enough content of both nutrients able to reduce Zn and Se deficiencies, obtaining a functional crop with added value for farmers. Therefore, the present study aims to evaluate the effect of the biofortification with Zn and Se, both individually and in combination, on the accumulation of these minerals in the edible parts of wheat (grain for humans and straw for animals), and on the yield and nutritional quality of such parts. Likewise, in the plots with Zn treatment in the soil, the evolution and permanence of Zn into soil was also evaluated to analyse its residual effect.

## 2. Results

### 2.1. Evolution of Soil Zn-DTPA in the Soil

The split-plot ANOVA performed to evaluate the residual effect of the Zn applications by analysing the concentration of Zn into the topsoil showed the main effects ‘sampling time’ (degree of freedom, df = 4, *F* value = 20.03, *p* < 0.001), ‘Zn application’ (df = 2, *F* value = 75.04, *p* < 0.001) and their interaction (df = 8, *F* value = 5.49, *p* < 0.001) to be all significant variables. The Zn-DTPA concentration increased significantly since its application. Such an increment was lower in 50SZn + 0FZn, with an average of 1.00 ± 0.10 mg Zn-DTPA ha^−1^, than in 50SZn + 8FZn, with 1.25 ± 0.12 mg Zn-DTPA ha^−1^, but only in the first measurement after its application. Afterwards, no significant variation was observed between both treatments, including Zn (Figure 1).

### 2.2. Zn and Se Concentrations and Contents in Straw and Its Bioavailability

Both the Zn concentration in the straw, and its bioavailability, measured through the molar ratio phytate/Zn, were significantly affected by the main effects ‘Study year’ and ‘Foliar application’, as well as by their interaction (Table 1). The foliar treatments containing Zn increase the Zn concentration in the straw almost 10-fold on average, although such differences took place mainly in the study year 2018/2019 (almost 16-fold) in comparison with 2017/2018 (more than 6-fold). The foliar Se application did not produce any effect on the concentration of Zn in the wheat straw (Figure 2). The molar ratio phytate/Zn in the straw was much lower in the treatments containing Zn, especially in the study year 2018/2019 (Table 2). When the total Zn content per ha was considered as response variable, only the foliar application had a significant influence (Table 1). In this case, the treatments containing Zn increased the total Zn content per ha about 8-fold (Figure 2).

Regarding Se, its concentration in the straw and the molar ratio Phytate/Se were only affected by the foliar treatment (Table 1). The treatments containing Se, regardless of the Zn application, produced on average almost a 3-fold increase in the Se concentration in comparison with the rest of treatments (Figure 3). The same pattern was observed for the molar ratio phytate/Se (Table 2). The Se content per ha in the straw, besides the foliar application, was also affected by the study year (Table 1), being 44% higher in 2017/2018 than in 2018/2019. Furthermore, the treatments containing Se produced the highest values of total Se content in the straw (Figure 3).

### 2.3. Zn and Se Concentrations and Contents in Grain and Its Bioavailability

The ‘soil Zn application’ and ‘foliar application’ as main effects, and the interaction ‘study year*foliar application’ significantly affected the Zn concentration in grain. The same pattern was observed for the Zn bioavailability (measured through the molar ratio phytate/Zn), excepting for the ‘soil Zn application’, which did not have significant influence (Table 1). Such as in the case of the straw, the foliar treatments containing Zn produced the highest Zn accumulation in the grain, but the magnitude of the increase was much lower in this case, at on average around 24% (Figure 2). In 2018/2019, those differences were even lower. The soil application of Zn also increased the Zn concentration in grain to around 9%. When considering the molar ratio phytate/Zn, again, the treatments containing Zn showed the lowest values, especially in 2017/2018 (Table 2). The total Zn content in grain was affected by the three main effects of study year (Y), soil Zn application (S) and foliar application (F), and by the interactions ‘Y*S’ and ‘Y*F’ (Table 1). The treatments including Zn, either soil or foliar applied, showed the highest values, but only in 2017/2018, the year with the highest total Zn content in the grain (Figure 2). The Se application did not have any effect on the Zn accumulation in the grain.

The concentration of Se in the grain, the total Se content and the molar ratio phytate/Se, were all affected by the main effects ‘study year’ and ‘foliar Zn application’ (Table 1). For Se concentration and total Se content, the pattern was almost the same in all the significant factors. The highest values were obtained in 2017/2018 in the treatments containing Se (Figure 3). The treatments containing Se produced the lowest values of the ratio Phytate/Se (Table 2).

### 2.4. Effect of Zn and Se Application on Grain and Straw Yield and Straw Nutritive Parameters

While in the case of the grain, yield was affected by the main effect ‘study year (Y)’ and by the interaction ‘Y*soil Zn application (S)’, the straw yield was significantly influenced by the main effects ‘Y’ and ‘S’ (Table 1). Grain yield was much higher in 2017/2018, the most humid growing season, than in 2018/2019 (almost 2-fold higher). The Zn application in soil caused an increase in the grain yield of around 10%, but only in 2017/2018. In 2018/2019, the soil Zn application did not have any effect in the grain yield (Figure 4). Straw was yielded more also in 2017/2018 than in 2018/2019 (more than 1.7-fold), and when Zn was applied to soil (more than 21%) in comparison with the non-fertilised control. In both cases, the Se application did not have any effect on the yield (Figure 4). The thousand grain weight and the hectolitre weight were only affected by the study year (Table 1). In both cases, values were higher in 2017/2018 than in 2018/2019: for thousand grain weight 35.8 ± 0.7% vs. 25.8 ± 0.4 g, and for hectolitre weight 79.7 ± 0.3% vs. 76.1 ± 0.4 kg hL^−1^, respectively.

Fibres (both neutral detergent, NDF, and acid detergent, ADF) and ashes were all only affected by the study year (Table 1). In all cases, the values were higher in 2017/2018 than in 208/2019 (for NDF: 72.8 ± 0.4% vs. 63.2 ± 0.4%; for ADF: 43.3 ± 0.3% vs. 34.2 ± 0.4%; for ashes: 1.9 ± 0.1% vs. 0.5 ± 0.0%, respectively. Lignin (LAD) was affected by the main effect ‘soil Zn application (S)’ and by the interaction ‘Study year*S’ (Table 1). In this case, the highest values were obtained when Zn was applied into soil, but only in 2017/2018 (Figure 5). Regarding the mineral status, the influence of the main factors studied and their interactions on their concentration and their bioavailability (measured through the molar ratio phytate/mineral) in the straw and grain can be observed in Table 1. Within those, the most significant results are shown in Table 3. While the study year influenced the Fe concentration and its bioavailability in both the straw and grain and the molar ratio phytate/Mg in the straw, the soil Zn application affected the Mg and Ca concentration in the straw and their bioavailability. The foliar application, although significant in some interactions with the study year, did not present a clear pattern (Table 3). In general, the highest concentration values for Mg, Ca and Fe were obtained in 2017/2018. In the case of Mg and Ca, such highest values were reached when soil Zn was not applied, and in the case of Fe, when foliar Zn and Se were applied in combination, but only in 2017/2018 (Table 3).

## 3. Discussion

The present study was designed to perform the soil Zn application only once at the beginning of the experiment, and with the minimum amount possible to reduce the total inputs that satisfy the crop requirements. In this research, after the soil application, the Zn-DTPA concentration in soil increased up to more than 1.00 mg kg^−1^, remaining always above 0.5 mg kg^−1^, which is a critical value to meet the crop needs [30]. This fact confirmed then the assumption that the used soil fertilisation rate was high enough to reach the values of available Zn into soil above the crop requirements in both the application year and at least in the following cropping year. This result agreed with the stated in previous studies [12,30], where an important Zn residual effect into soil after a Zn sulfate fertiliser application was reported. However, more years are needed to establish, in semi-arid conditions, what is the duration of the effect of this application.

Without biofortification, the values of Zn and Se concentration in the straw were on average 8.5 mg kg^−1^ and 33.0 μg kg^−1^, respectively, while in the grain were 33.1 mg Zn kg^−1^ and 26.3 μg Se kg^−1^. Considering that the required amount of Zn and Se by humans is about 15 mg Zn day^−1^ and 55 μg Se day^−1^, respectively [33,34], and that livestock requires about 35 mg Zn kg^−1^ and 0.1–0.5 mg Se kg^−1^ feed DM, respectively [35], without biofortification, the levels reached in the different parts of the wheat plant might be under these values. This fact supports the idea that low levels of Zn and Se into soil, such as it was in this case, might produce plants with inadequate Zn and Se concentration in their edible parts to accomplish the required necessities in humans and livestock. Under these soil conditions, the implementation of strategies like agronomic biofortification, which allow alleviating such deficiencies, might make much more sense than in other situations. To get a general application, while the obtained plant-derived products do not reach higher prices because of the Zn and Se enrichment, public policies should fund to farmers the extra costs generated by the application.

Regarding the situation without fortification, two results are interesting to be remarked. The first is that while for Se, the concentration was quite similar in the straw and in the grain, for Zn, it was much higher in the grain than in the straw. Because Zn is an essential nutrient for plants involved in many physiological and metabolic processes [36], the Zn accumulated in senescent tissues might be transported to younger sinks still in development to be again used in the cellular activity of the novel part. This might be supported by Longnecker and Robson [37], who indicated that Zn concentrations are usually higher in growing tissues than in those that are mature. In the case of Se, because it is essential for mammals [38,39], but not for plants [40], although different positive effects have been reported, such as an increase on the chlorophyll content accumulation on the leaves [41] with an improvement in the photosynthetic system [42], alleviated adverse effects of drought stress in different species [43], maintaining under heat or drought stress and grain yield in cereals [44]. The second aspect to be remarked is that for Se, its concentration was higher in the most humid year (2017/2018) in the grain, but for Zn the effect was opposite, i.e., the highest values were obtained in the growing season with the lowest rainfall (2018/2019) in the straw (Figure 6). This apparently contradictory result can be explained by a dilution effect, as a result of the different yield obtained. When the total mineral content (multiplying the mineral concentration by the yield) is considered to take into account this effect, in both cases, for straw and grain, and for both micronutrients, the values obtained in the most humid year (2017/2018) were almost 2-fold of those in 2018/2019. Therefore, as stated previously [45,46], water availability might enormously favour the uptake of these micronutrients.

When biofortification was performed, the best treatment to increase both Zn and Se concentration in both straw and grain was the combined foliar application of 8 kg zinc sulfate ha^−1^ and 10 g sodium selenate ha^−1^. With this application, the values of Zn and Se concentration in the straw were on average 98.8 mg Zn kg^−1^ (an increase of 12.3-fold in comparison with that of controls) and 87.6 μg Se kg^−1^ (2.7-fold that of controls), respectively, while in the grain were 46.4 mg Zn kg^−1^ (1.32-fold that of controls) and 124.9 μg Se kg^−1^ (4.3-fold that of controls), respectively. According to these results, the importance of the biofortification effect, although always effective, was quite inconsistent, depending on the plant tissue analysed and the mineral considered. To explain this inconsistency, data should be analysed considering each study year separately. 

In 2018/2019, a year with an unusual very low precipitation (Figure 6), the foliar application affected at a lower extent the Zn concentration in grain. However, such an application had a very important influence in the Zn concentration in the straw, resulting in values 16.7-fold higher when both nutrients were applied in comparison with the non-fertilised controls. Therefore, under this situation, it seemed that the Zn absorbed by leaves after application was not so effectively transported to the grain, remaining mostly in the foliar system. This might be supported by the fact that phloem transport, main Zn fed source for young sink tissues such as developing grains [47], it is known to fail, or at least decrease, during drought [48]. In 2017/2018, the Zn concentration increased more importantly in the straw than in the grain (6.5-fold vs. 1.5-fold, respectively). Although the rainfall was higher in this year, it might not be still enough for an adequate transport to the grain. Therefore, under the semiarid Mediterranean climate, characterised by scarce and very irregular precipitations, the effectiveness of the Zn biofortification might be clear and positively linked with the amount of rainfall. Nevertheless, further studies, including a higher number of study years or designing specific experiments with different water regimes, should be performed in order to clarify the exact influence of the water availability in the efficiency of the Zn biofortification.

Regarding the Se accumulation after its foliar application, while in the straw the importance of the increase was quite similar regardless of the study year (2.1-fold in 2017/2018 and 2.9 in 2018/2019), in the grain the influence of the year was determinant. In the driest year, 2018/2019, the foliar application increased the Se concentration in grain around 2.8-fold, while in the most humid year, 2017/2018, it increased almost 5.4-fold. Again, as in the case of Zn, precipitation seems to substantively affect the efficiency of the Se biofortification, such as it was also found in previous studies on bread-making wheat [9,11,29]. The quite greater plant vegetative growth in 2017/2018, figured out after observing the greater straw yield of this year, and an increased opening of the leaf stomata, the main route for foliar nutrients entrance into the plant [49], as a consequence of the higher rainfall, might explain this highest efficiency. In any case, even in the most effective case, i.e., in the year 2017/2018 when Zn and Se were simultaneously applied, the Se concentration in grain was quite lower than that obtained previously for bread-making wheat in a very close area [9,11,29], which accounted for almost 800 μg kg^−1^ (vs. the 182 μg kg^−1^ of the present study). The higher amount of rainfall and the higher initial total Se in the topsoil (6 μg extractable Se kg^−1^ vs. 1.27 μg extractable Se kg^−1^) of that study could explain the observed differences. Another factor which has been regarded to affect the Se uptake in bread-making wheat grain is the rain fallen during the days before the Se application [29]. There it was found that the lower the precipitation in this period, the higher the Se accumulation in grain. Considering all of these aspects, under the semiarid conditions of the Mediterranean climate, special attention should be paid to rainfall, especially in the days prior to the fertiliser application, in order to maximise the success of the combined biofortification with Zn and Se in the driest years.

In terms of biofortification, not only the concentration of the target micronutrients (Zn and Se in this case) is important, which increased by the way to values close to the recommended under the studied conditions, but also how they are bioavailable for the organism. In this regard, phytates are phosphorous-containing compounds that reduce the nutrient absorption, especially for Ca, Fe, Mg and Zn [50]. Considering that a phytate:Zn molar ratio lower than 15 is associated with high Zn bioavailability [51], foliar treatments containing Zn, especially for the straw, caused a decrease in such values below this threshold. Furthermore, biofortification is important to be analysed in terms of productivity and nutritional quality of the edible parts. As the application of Zn in deficient soil increased photosynthesis by increasing chlorophyll a and b concentrations, transpiration and stomatal conductance rate [52,53], a yield increase is expected. In this experiment, the combined soil fertilisation with 50 kg Zn ha^−1^ and the foliar application of 8 kg Zn ha^−1^ + 10 g Se ha^−1^ increased on average the yield of grain, the main product of this crop, by almost 7%. Although less important in the global farm incomes, the straw yield also increased by around 26% after that combined application. Both increases, especially that of the grain yield, might give solid arguments able to persuade farmers to implement these programs, besides the increase in the Zn and Se concentration in their products which, although beneficial for society and stockbreeders, are still not compensated in their selling prices. Future trials should aim at facilitating and reducing costs to make agronomic biofortification with Se and Zn even more attractive for farmers. A good example is that reported by Wang et al. [54], who combined Zn application plus pesticide, showing it as a cost-effective ready-to-use strategy to fight human Zn deficiency in wheat-dominated regions around the world.

The rest of parameters analysed, such as straw fibres, lignin, or mineral status (Mg, Ca, Fe), resulted in either unaffected or very few affected by the biofortification with Zn and Se. Once again, this is a good result to try to successfully implement these programs among farmers, as its application might not be detrimental for the quality of their productions.

## 4. Materials and Methods

### 4.1. Site, Experimental Design and Crop Management

The field experiment was conducted in Badajoz, southern Spain (38°54′ N, 6°44′ W, 186 m above sea level), in a Xerofluvent soil, according to Soil Taxonomy, under rainfed Mediterranean conditions in 2017/2018 and 2018/2019 growing seasons. Weather-related parameters for this area for the concerned years, as well as for the average year obtained from a 30-year period, are shown in Figure 6. All climate data were taken from a weather station located at the study site. In the first study year, rainfall was like the average year but much higher (40%) than in 2018/2019. The months of March and April were exceptionally rainy, with ~175 and ~80 mm, respectively, which is much higher than in the average year. The second year was extraordinarily dry, with a very different seasonal distribution, being autumn the wettest season, to the extent that it caused a two-month delay in sowing compared to the first year, and with important drought periods between February and March and May and June.

The experiment was designed as a split-split plot arrangement with four replications randomly distributed. The main plots were the study year (2017/2018 and 2018/2019), subplots were Zn soil application (without any application [0SZn] and with a soil application of 50 kg ZnSO_4_-7H_2_O ha^−1^ [50SZn]) (equivalent to 11.4 kg of Zn ha^−1^) and sub-subplots were foliar application with four treatments (without any application [0F]; two foliar applications of 4 kg ZnSO_4_-7H_2_O ha^−1^ each (equivalent to 0.91 kg of Zn ha^−1^) at the start of flowering and two weeks later [8FZn]: a foliar application of 10 g Se ha^−1^ as Na_2_SeO_4_ (equivalent to 24 g ha^−1^ of Na_2_SeO_4_) at the start of flowering [10FSe]: a combination of 8FZn and 10FSe [8FZn + 10FSe]). The crop area for each treatment was 15 m^2^ (3 m × 5 m). Zinc soil treatment was only made at the beginning of the first season, in October 2017, before the sowing, sprayed as a solid in the soil surface and incorporated into the soil by tillage. 

The foliar Zn application treatment consisted of two times of foliar Zn application at the start of flowering, and it was repeated two weeks later as described by Gomez-Coronado et al. [12]. At each time, 0.5% (*w*/*v*) of aqueous solution of ZnSO_4_·7H_2_O ha^−1^ with 800 L per hectare were sprayed until most of the leaves were covered at the very late afternoon to avoid burning in plants. For the Se treatment 10 g of Se applied as Na_2_SeO_4_ ha^−1^ was diluted in 800 L H_2_O ha^−1^ to obtain a 0.003% (*w*/*v*) solution, and applied as in the case of foliar Zn, as described by Poblaciones et al. [9].

The bread wheat (*Triticum aestivum* L.) cultivar used was “Antequera”. Conventional tillage treatment was used to prepare a proper seedbed before sowing. The sowing was in late October in the first year (2017) and late December in 2018 (due to the intense rainfall in autumn in the second year). The sowing rate was of 350 seeds m^2^; each plot had six rows of 20 cm apart. A N-P-K fertiliser (15-15-15) was applied before sowing at a 200 kg ha^−1^ dose in all plots. Weed control was carried out by applying Trigonil (concentrated in suspension to 400 g L^−1^ of chlortoluron and 25 g L^−1^ of diflufenican) in the sowing. 

### 4.2. Soil Analysis

To characterise the experimental soil, four representative soil samples of 30 cm depth were taken in September 2017 from the experimental site. Soil samples were air dried and sieved to <2 mm using a roller mil. Texture was clay loam, determined gravimetrically; soil pH was slightly acid, with 6.4 ± 0.2 (mean ± standard error) using a calibrated pH meter (ratio, 10-g soil:25-mL deionised H_2_O), soil organic matter was very low with 1.31 ± 0.09% determined by oxidation with potassium dichromate [55], total N was medium with 0.12 ± 0.007% [56], P Olsen with 4.9 ± 0.05 g P kg^−1^ was low, measured following the Olsen procedure, and assimilable K was low with 321 ± 8 mg kg ^−1^, determined with ammonium acetate (1N) and quantified by atomic absorption spectrophotometry.

Soils contained low concentrations of Ca with 1248 ± 134 mg kg^−1^, medium of Mg with 1455 ± 145 mg kg^−1^. They were extracted according to the method of [57] by extraction with DTPA (diethylenetriamine pentaacetic acid) and measured by inductively coupled plasma optical emission spectroscopy (ICP-MS) (Agilent 7500ce, Agilent Technologies, Palo Alto, CA, USA). Extractable Se was very low (with 1.27 ± 0.01 µg Se kg^−1^) determined by using KH_2_PO_4_ (0.016 mmol L^−1^, pH 4.8) at a ratio of 10 g dry weight soil: 30 mL KH_2_PO_4_ (*w*/*v*) [58]. The Se concentration in the extracts was determined by ICP-MS, as described below. All the results were reported on a dry weight basis.

To evaluate the residual effect along the experiment of the Zn soil treatments, i.e., NoSZn, 50SZn + 0F and 50SZn + 8FZn, four sampling, as well as the initial sampling, was taken in January in both study years and before each harvest (therefore in September 2017, January 2018, May 2018, January 2019 and May 2019). Zinc-DTPA were determined in each moment.

### 4.3. Plant Analysis

Harvesting was done at maturity in early July. Straw and grain samples were thoroughly washed with tap water, and then with distilled water to avoid the eventual presence of residues from foliar applications. Afterwards, samples were dried at 70 °C until constant weight, and their dry matter yield was then recorded. Thousand grain weight and hectolitre weight were determined from the grain samples. Official procedures [59] were followed to determine neutral detergent fiber (NDF), acid detergent fiber (ADF) and acid detergent lignin (ADL) by means of a fiber analyser (ANKOM8–98, ANKOM Technology, Macedon, NY, USA). Total ash content was determined by ignition of the sample in a muffle furnace at 600 °C, as is indicated in the official procedure [59]. Total straw and grain mineral concentration (Ca, Fe, Mg, Se and Zn) were determined as follows: straw and grain were finely grounded (<0.45 mm) using an agate ball mill (Retch PM 400 mill); a 1 g was digested with ultra-pure concentrated nitric acid (2 mL) and 30% *w*/*v* hydrogen peroxide (2 mL) using a closed-vessel microwave digestion protocol (Mars X, CEM Corp, Matthews, NC, USA) and diluted to 25 mL with ultra-purified water [60]. Sample vessels were thoroughly washed with acid before use. For quality assurance, a blank and a standard reference material (tomato leaf, NIST 1573a) were included in each batch of samples. The digested was determined by ICP-MS. The studied mineral recovery was 95%, compared with certified reference material (CRM) values. To consider the dilution effect in Zn and Se caused by the different straw and grain yield between growing seasons, total nutrient uptake was calculated multiplying grain yield by the total Zn and Se concentration in grain. On the other hand, phytate concentration was estimated by means of phytic acid, whose determination is based on precipitation of ferric phytate and measurement of iron (Fe) remaining in the supernatant [61]. Phytate was extracted from about 0.2 g of ground straw or grain in 10 mL of 0.2 M HCl (pH 0.3) after shaking for 2 h. One ml of supernatant was treated with 2 mL of ferric solution (NH_4_Fe(SO_4_)_2_·12 H_2_O) in a boiling water bath for 30 min. After cooling, samples were centrifuged, and 1 mL of supernatant was treated with 1.5 mL of 0.064 M bipyridine (2-pyridin-2-ylpyridine, C_10_H_8_N_2_) to measure Fe. After mixing, the solution was incubated for 10 min at room temperature, and the light absorbance was measured with a spectrophotometer at 419 nm. Finally, the molar ratio between phytate and Ca, Fe, Mg, Se and Zn was calculated to estimate the bioavailability of those nutrients.

### 4.4. Statistical Analysis

The evolution of the Zn soil treatments on the soil Zn-DTPA was evaluated by a split-plot analysis of variance (ANOVA). The main-plot factor was “sample time” (before starting, in January and harvest of 2018, and in January and harvest of 2019), the subplot factor “Zn application” (0SZn + 0F, 50SZn + 0F, and 50SZn + 8FZn), and its interaction in the model. 

Data of mineral concentration (Ca, Fe, Mg, Se and Zn) and phytate/mineral molar ratios in straw and grain, as well as straw and grain yield, thousand grain weight and hectolitre weight, and nutritive value parameters of the straw were subjected to split-split-plot ANOVAs, including the main-plot factor ‘study year’ (2017/2018 and 2018/2019), the subplot factor ‘soil Zn application’ (0SZn and 50SZn), the sub-subplot factor ‘foliar application’ (0F, 8FZn, 10FSe and 8FZn + 10FSe), and their interactions in the model. When significant differences were found in ANOVA, means were compared using Fisher’s protected least significant difference (LSD) test at *p* ≤ 0.05. All these analyses were performed with the Statistix v. 8.10 package (Analytical Software, Tallahessee, FL, USA). In order to normalise the variable distribution, as well as to stabilise the variance of residues, the transformation Ln(x + 1) was performed for the concentration of Se in grain, total Se content in grain and straw, the molar ratio phytate/Se and the ash content in the straw.

## 5. Conclusions

The results presented here showed bread-making wheat to be a very suitable crop to be used in biofortification programs with Zn and Se, as it was able to substantially accumulate the Zn and Se applied in combination in the edible parts, to alleviate their deficiency in people when used as staple food, or in livestock when using its straw. In fact, the combined foliar application of Zn and Se increased in straw, 12.3- in Zn and 2.7-fold in Se, and grain, 1.3- and 4.3-fold, respectively. However, the efficiency of the uptake and later accumulation was highly affected by the rainfall. Thus, in Mediterranean climates, characterised by irregularity in precipitations, the years with extensive drought periods could account for lower values, especially in the grain. In addition to the higher Zn and Se concentration in the edible parts, the application of 50 kg Zn ha^−1^ produced on average an increase of around 7% in the grain yield, and 26% in the straw yield, with the remaining productive and nutritive quality parameters almost unaffected.

## Figures and Tables

**Figure 1 plants-10-01209-f001:**
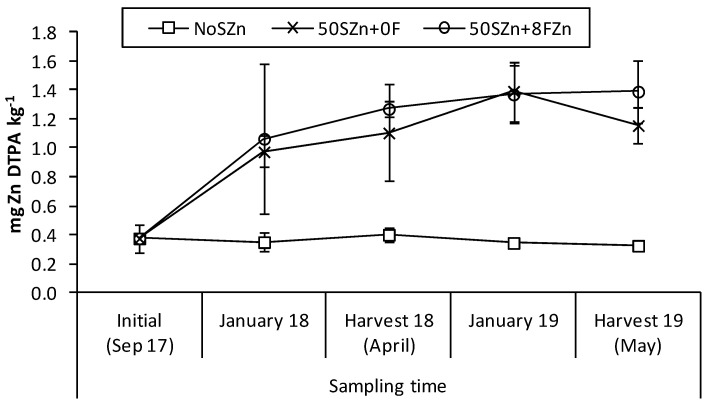
Zn DTPA concentration into topsoil of the study area as affected by the interaction ‘sampling time (five times) * Zn application (3 treatments: NoSZn, 50SZn + 0F and 50SZn + 8FZn)’. Error bars indicate standard error (*n* = 3). Different letters mean significant differences between means according to LSD test (*p* ≤ 0.05).

**Figure 2 plants-10-01209-f002:**
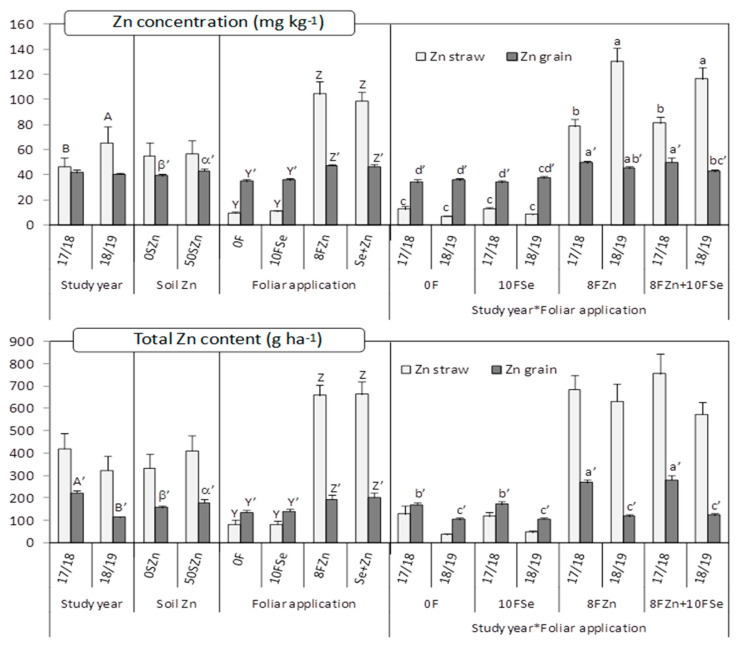
Concentration of Zn and total Zn content in the straw and grain as affected by the main effects ‘Study year (Y)’, ‘Soil Zn application (S)’, ‘Foliar application (F)’, and by the interaction ‘Y*F’. Charts indicate means (for straw *n* = 3; for grain *n* = 4) and error bars indicate standard error. Within each factor and plant part, different letters mean significant differences between means according to LSD test (*p* ≤ 0.05). To make the differences clearer, a different set of letters was assigned to each factor and plant part (lowercase letters for ‘Y*F’, Greek letters for ‘S’, uppercase letters [A, B] for ‘Y’ and uppercase letters [Z, Y] for ‘F’). Letters follow by apostrophe (‘) for grain.

**Figure 3 plants-10-01209-f003:**
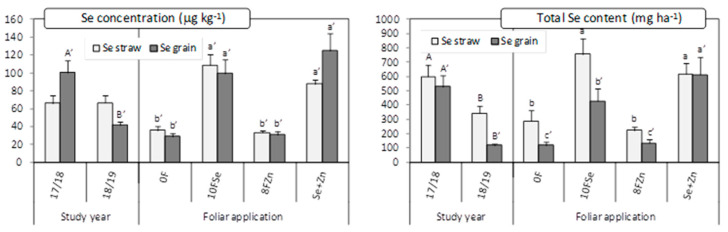
Concentration of Se and total Se content in the straw and grain as affected by the main effects ‘Study year (Y)’ and ‘Foliar application (F)’. Charts indicate means (for straw *n* = 3; for grain *n* = 4) and error bars indicate standard error. Within each factor and plant part, different letters mean significant differences between means according to LSD test (*p* ≤ 0.05). To make the differences clearer, a different set of letters was assigned to each factor and plant part (lowercase letters for ‘F’, uppercase letters for ‘Y’). Letters follow by apostrophe (‘) for grain. Although the LSD test was performed on the transformed variable, back-transformed values are represented to ease interpretation.

**Figure 4 plants-10-01209-f004:**
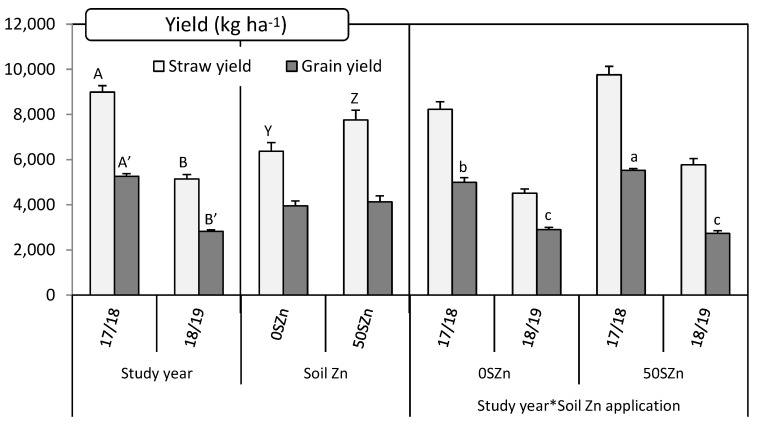
Influence of the main effects ‘Study year (Y)’, ‘Soil Zn application (S)’ and their interaction ‘Y*S’ on both straw and grain yield. Charts indicate means (*n* = 4) and error bars indicate standard error. Within each parameter and factor, different letters mean significant differences between means according to LSD test (*p* ≤ 0.05). To make the differences clearer, a different set of letters was assigned to each factor (lowercase letters (a, b, c) for the interaction, uppercase letters [A, B] for ‘Y’ and uppercase letters (Z, Y) for ‘S’. Letters follow by apostrophe (‘) for grain.

**Figure 5 plants-10-01209-f005:**
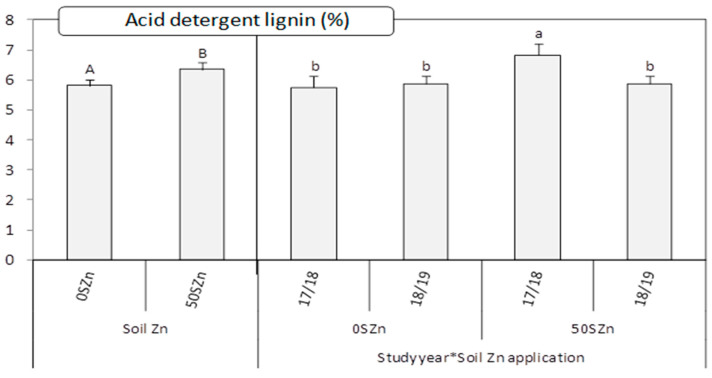
Acid detergent lignin for straw as affected by the main effect ‘Soil Zn application (S)’ and by the interaction ‘Study year (Y)*S’. Charts indicate means (*n* = 4) and error bars indicate standard error. Within each factor, different letters mean significant differences between means according to LSD test (*p* ≤ 0.05). To make the differences clearer, a different set of letters was assigned to each factor (lowercase letters for ‘Y*S’ and uppercase letters for ‘S’).

**Figure 6 plants-10-01209-f006:**
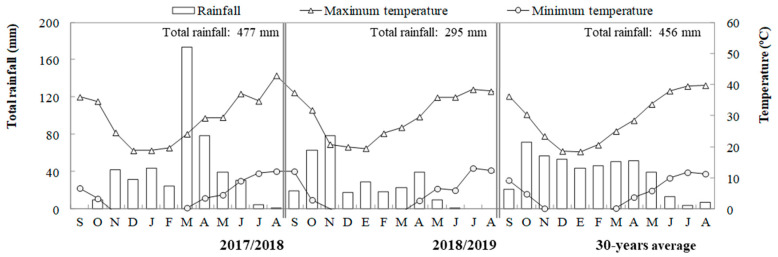
Monthly and annual rainfall and mean maximum and minimum temperatures in 2017/2018, 2018/2019 and in an average year from a 30-year period in Badajoz (Spain).

**Table 1 plants-10-01209-t001:** Summary of the split-split-plot ANOVAs showing the effect of the main-plot factor (year), subplot factor (Zn soil application), sub-subplot factor (foliar application) and their interactions on each parameter evaluated in straw and grain. DF, degree of freedom; F values, including the level of significance (* *p* ≤ 0.05, ** *p* ≤ 0.01, *** *p* ≤ 0.001) are shown in the rest of the rows.

	Part	Year (Y)	Zn Soil Applic. (S)	Foliar Applic. (F)	Y*S	Y*F	S*F	Y*S*F
DF		1	1	3	1	3	3	3
Zn (mg kg^−1^)	Straw ^1^	35.92 *	0.23	185.9 ***	1.25	13.70 ***	0.26	1.33
	Grain	1.36	7.30 *	26.71 ***	0.00	3.75 *	0.10	0.49
Se (μg kg^−1^)	Straw ^1^	0.07	0.06	27.78 ***	0.04	0.22	0.34	0.37
	Grain ^2^	75.01 **	0.33	55.93 ***	0.17	2.36	0.05	0.70
TZn (g kg^−1^)	Straw ^1^	5.39	6.31	100.6 ***	1.48	0.69	0.53	1.09
	Grain	101.2 **	24.01 **	25.92 ***	15.83 **	13.78 ***	0.50	0.26
TSe (mg kg^−1^)	Straw ^1,2^	29.45 *	1.04	47.86 ***	0.03	0.91	0.81	0.85
	Grain ^2^	365.4 ***	0.50	53.62 ***	1.23	2.34	0.07	0.92
Phytic acid (g kg^−1^)	Straw ^1^	1.04	0.30	0.24	0.11	2.18	1.26	1.47
	Grain	0.04	3.09	0.33	0.93	0.45	0.20	0.14
Ph/Zn	Straw ^1^	49.29 *	1.62	96.10 ***	0.46	6.61 **	0.90	1.09
	Grain	0.03	5.09	39.30 ***	0.02	4.29 *	0.68	0.61
Ph/Se	Straw ^1^	0.05	0.04	57.46 ***	0.02	0.85	0.10	0.30
	Grain ^2^	61.83 **	0.24	54.70 ***	0.13	1.68	0.06	0.79
Yield (kg ha^−1^)	Straw	195.4 ***	28.55 **	0.64	0.27	1.58	1.70	0.78
	Grain	394.7 ***	2.23	1.70	8.01 **	2.19	1.42	0.55
1000 gw (g)	Grain	60.75 **	0.16	2.72	0.30	1.24	0.08	1.67
Hect. weight (kg hL^−1^)	Grain	31.38 *	0.29	2.34	3.16	0.91	0.82	0.78
NDF (%)	Straw	1766 ***	1.46	0.88	0.18	0.42	0.60	0.12
ADF (%)	Straw	937.1 ***	2.19	0.23	0.69	0.24	0.63	0.09
ADL (%)	Straw	2.14	8.13 *	1.22	7.65*	1.52	0.88	1.53
Ashes (%)	Straw ^2^	102.1 **	0.07	0.32	3.35	0.38	1.74	1.09
Mg (mg kg^−1^)	Straw ^1^	16.92	14.71*	0.56	3.77	0.48	0.87	0.79
	Grain	0.35	0.02	1.03	1.98	1.58	0.58	1.35
Ca (mg kg^−1^)	Straw ^1^	8.25	85.14 ***	0.44	71.44 **	0.37	1.05	1.58
	Grain	2.73	0.49	1.17	1.94	2.10	1.07	2.33
Fe (mg kg^−1^)	Straw ^1^	101.8 **	1.18	0.92	0.17	1.15	2.41	0.41
	Grain	14.56 *	0.40	0.69	5.87	3.85 *	0.89	0.49
Ph/Mg	Straw ^1^	18.60 *	10.54 *	0.96	1.41	1.09	0.51	0.67
	Grain	0.00	0.18	1.31	3.00	1.82	1.45	1.57
Ph/Ca	Straw ^1^	9.94	18.00 *	0.93	15.13 *	1.53	0.95	1.80
	Grain	0.14	1.10	1.57	2.33	2.08	1.52	2.55
Ph/Fe	Straw ^1^	403.9 **	2.34	1.86	0.12	1.71	2.45	0.30
	Grain	25.57 *	0.31	0.54	9.97*	4.21*	0.90	0.51

TZn: total Zn content = Zn*yield; TSe: total Se content = Se*yield; Yield: grain yield; 1000 gw: thousand grain weight; NDF: neutral detergent fiber; ADF: acid detergent fiber; ADL: acid detergent lignin; Ph/mineral: molar ratio Phytate/each mineral.^1^ In these parameters: *n* = 3; in the rest: *n* = 4. ^2^ These parameters were transformed by following: Ln(x + 1).

**Table 2 plants-10-01209-t002:** Molar ratio phytate: mineral (Zn and Se) in the straw and grain, expressed as mean value ± standard error (*n* = 3 for straw and *n* = 4 for grain) as affected by the main effects ‘Study year (Y)’, ‘Foliar application (F)’ (in bold) and by their interaction ‘Y*F’.

		Factor	Treatment	Study Year		
				2017/2018	2018/2019	Average
Phytate:Zn	Straw	Foliar application	0F	52.9 ± 10.1 bc	84.5 ± 8.4 a	68.7 ± 7.8 Z
			10FSe	45.0 ± 5.5 c	66.5 ± 4.7 b	55.8 ± 4.7 Y
			8FZn	7.2 ± 0.5 d	4.4 ± 0.3 d	5.8 ± 0.5 X
			8FZn + 10FSe	7.1 ± 0.5 d	4.9 ± 0.3 d	6.0 ± 0.4 X
			Average	28.0 ± 5.1 B	40.1 ± 8.0 A	
	Grain	Foliar application	0F	17.3 ± 1.0 a	16.4 ± 0.8 ab	16.9 ± 0.6 Z
			10FSe	17.1 ± 0.8 ab	15.5 ± 0.4 b	16.3 ± 0.5 Z
			8FZn	11.7 ± 0.3 d	12.9 ± 0.5 cd	12.3 ± 0.3 Y
			8FZn + 10FSe	12.0 ± 0.7 d	13.6 ± 0.4 c	12.8 ± 0.4 Y
			Average	14.5 ± 0.6	14.6 ± 0.4	
Phytate:Se	Straw	Foliar application	0F	19.9 ± 2.9	22.5 ± 1.8	21.2 ± 1.6 Z
			10FSe	8.1 ± 1.8	6.2 ± 0.5	7.2 ± 0.9 Y
			8FZn	20.9 ± 1.3	21.8 ± 3.1	21.4 ± 1.5 Z
			8FZn + 10FSe	8.2 ± 0.7	7.7 ± 0.5	8.0 ± 0.4 Y
			Average	14.3 ± 1.5	14.5 ± 1.8	
	Grain	Foliar application	0F	24. 0 ± 3.1	35.0 ± 5.7	29.5 ± 3.4 Z
			10FSe	5.5 ± 0.8	14.0 ± 1.3	9.8 ± 1.4 Y
			8FZn	20.4 ± 4.0	37.3 ± 5.5	28.9 ± 3.9 Z
			8FZn + 10FSe	4.5 ± 0.8	11.0 ± 1.2	7.7 ± 1.1 Y
			Average	13.6 ± 2.0 B	24.3 ± 2.8 A	

Within each parameter and factor, different letters mean significant differences between means according to LSD test (*p* ≤ 0.05). If letters do not appear, this factor did not have a significant effect according to split-split-plot ANOVA. To make the differences clearer, a different set of letters was assigned to each factor (lowercase letters [a, b, c, d] for ‘Y*F’, uppercase letters [Z, Y, X] for ‘F’ and uppercase letters [A, B] for ‘Y’.

**Table 3 plants-10-01209-t003:** Concentration of Mg, Ca and Fe, and their molar ratio phytate:mineral in either the straw or grain, expressed as mean value ± standard error (*n* = 3 for straw and = 4 for grain) as affected by the main effects ‘Study year (Y)’, ‘Zn soil application (S)’ and/or ‘Foliar application (F)’ (in bold) and by the interactions ‘Y*S’ and/or ‘Y*F’.

		Factor	Treatment	Study Year		
				2017/2018	2018/2019	Average
Straw	Mg (g kg^−1^)	Soil Zn application	0SZn	1.05 ± 0.05	0.77 ± 0.03	0.91 ± 0.04 Z
			50SZn	0.85 ± 0.03	0.71 ± 0.02	0.78 ± 0.02 Y
			Average	0.95 ± 0.03	0.74 ± 0.02	
	Ph/Mg	Soil Zn application	0SZn	0.20 ± 0.01	0.27 ± 0.01	0.24 ± 0.01 Y
			50SZn	0.25 ± 0.01	0.30 ± 0.01	0.27 ± 0.01 Z
			Average	0.22 ± 0.01 B	0.28 ± 0.01 A	
	Ca (g kg^−1^)	Soil Zn application	0SZn	3.73 ± 0.15 z	2.67 ± 0.06 x	3.20 ± 0.14 Z
			50SZn	3.05 ± 0.13 y	2.64 ± 0.08 x	2.84 ± 0.09 Y
			Average	3.39 ± 0.12	2.65 ± 0.05	
	Ph/Ca	Soil Zn application	0SZn	0.09 ± 0.00 x	0.13 ± 0.00 z	0.11 ± 0.00 Y
			50SZn	0.12 ± 0.00 y	0.13 ± 0.00 z	0.12 ± 0.00 Z
			Average	0.10 ± 0.00	0.13 ± 0.00	
Grain	Fe (mg kg^−1^)	Foliar application	0F	36.3 ± 0.9 bc	35.0 ± 2.6 bcd	35.6 ± 1.3
			10FSe	39.3 ± 1.9 ab	34.7 ± 2.9 bcd	37.0 ± 1.7
			8FZn	37.8 ± 1.8 ab	32.0 ± 0.6 cd	34.9 ± 1.2
			8FZn + 10FSe	42.3 ± 2.9 a	30.6 ± 0.9 d	36.4 ± 2.1
			Average	38.9 ± 1.0 A	33.1 ± 1.0 B	
	Ph/Fe	Soil Zn application	0SZn	1.24 ± 0.02 x	1.65 ± 0.02 z	1.44 ± 0.04
			50SZn	1.38 ± 0.05 y	1.44 ± 0.05 y	1.41 ± 0.04
		Foliar application	0F	1.38 ± 0.04 cd	1.47 ± 0.09 bc	1.43 ± 0.05
			10FSe	1.29 ± 0.06 de	1.49 ± 0.09 abc	1.39 ± 0.06
			8FZn	1.34 ± 0.07 cde	1.56 ± 0.03 ab	1.45 ± 0.05
			8FZn + 10FSe	1.21 ± 0.08 e	1.64 ± 0.05 a	1.43 ± 0.07
			Average	1.31 ± 0.03 B	1.54 ± 0.03 A	

Within each parameter and factor, different letters mean significant differences between means, according to LSD test (*p* ≤ 0.05). If letters do not appear, this factor did not have a significant effect according to split-split-plot ANOVA. To make the differences clearer, a different set of letters was assigned to each factor (lowercase letters [a, b, c, d, e] for ‘Y*F’, lowercase letters (z, y, x) for ‘Y*S’, uppercase letters [Z, Y] for ‘S’ and uppercase letters [A, B] for ‘Y’.

## Data Availability

Data sharing is not applicable to this article.

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
