# Peer review of "Combined Selenium and Zinc Biofortification of Bread-Making Wheat under Mediterranean Conditions"

_plants, 2021, doi:10.3390/plants10061209_

Round 1

Reviewer 1 Report

 Review of combined Se and Zn-plants-1240696

 The manuscript is well-written and provides interesting results and information. My special comments are as follows.

 L16. Replace nil with no or 0 Zn

L17. Replace nil with no or 0 Se

L30. Ton is an old term, replace it with Mg (megagram) throughout the manuscript.

L40. Expand it to full when use first time

L41. It can not be g Se kg-1 but µg, please correct it.

L105-109.  Could manyfold increase in Zn in straw be related to residual Zn remaining on the leaf after spraying. How can you ascertain it that it is not the case?

 Table 1. State at the bottom of the table what FND, FAD, LAD stand for

 L218.  Replace worthy with noteworthy

L303. Replace cheapen with reduced

L337-340. Could lower yield in 2018/19 cause lower dilution and hence higher concentration.

 L435. Give classification reference (e.g. Soil Taxonomy or FAO)

L450: Replace repetitions with replications

L457-58. State if Zn was applied in solid or solution form to make uniform application

L518.  State in full CRM (e.g. certified reference material) when used first time.

Author Response

Our comments can be found in the attachment

Reviewer 2 Report

This proposed manuscript intended to improve Selenium and Zn concentrations in the grain of the bread making wheat (Triticum aestivum L. - Antequera). It was combined foliar Se (selenate form) and soil and foliar Zn applications, in a two-year field experiment under Mediterranean conditions.  The combined Zn and Se application also increased on average the yield of grain, by almost 26%.

Concerning to the quality of the paper: The title of this paper clearly reflects its content. The objectives are clear and appropriate in the view of this subject matter. The results and discussion of this study has a sound English. In general, references are recent and adequate. So, it is a very interesting study, and I recommend its publication in the present form.

Author Response

The authors would like to thank the reviewer for his kind comments. Due to the comments made by the other two reviewers, we really think that the paper has been improved in some of the aspects mainly in the introduction and discussion making it more comprehensible and more readable.

Reviewer 3 Report

The article deals with topics of great importance to the modern world, concerning the shortages of ingredients in the human diet. The authors discuss the effectiveness of wheat biofortification into zinc and selenium depending on many factors such as weather conditions in the growing season, soil fertilization with zinc, foliar fortification with zinc and/or selenium. They also discuss the bioavailability of elements from the obtained crop. The conclusions are supported by in-depth statistical analysis.

However, there are some shortcomings in the article:

  1. in the case of this type of research, describing the cultivation trial only in chapter 4 is cumbersome. It is impossible to understand chapters 2 and 3 without knowing the basics of experience,

line 64 the error in the chemical compound entry. should be ZnSO4‧H2O,

table 1 the units in which the presented parameters are expressed are not given

table 1 the last character in the table header is omitted. Is 'Y * S *' should be 'Y * S * F'

table 3 in the Ph / Mg results, the average results for the 2017/18 and 2018/19 season are  the same, but the authors marked them as statistically different. If it's not a mistake, I suggests giving the results with greater accuracy

figure 6. X axis signatures (months) is neither intuitive nor described anywhere in the text. Please specify this information.

figure 6 the line of maximum and minimum average temperature should not be connected with the corresponding graphs for the seasons in which the experiment was conducted. The results presented in this way suggest the continuity of the presented results, and it is not

line 454-455 why the authors give the dose of Zn as ZnSO4‧H2O (a chemical compound) but selenium as calculated for the Se (element) alone. This is an inconsistency. I am asking for unification.

Chapter 4.2 the authors give the content of elements in the soil in meq / 100g. These are old units that are rarely used today. I suggest presenting the results in SI units.

Author Response

(The authors gave the same response as above.)

Reviewer 4 Report

Authors, in an interesting and very comprehensive study, summarized the roles of Selenium and Zinc in wheat nutrition, for higher plant production and improved quality. The authors evaluated the efficacy of combined foliar Se and Zn fertilization in bread-making wheat as foliar Se and soil and foliar Zn applications applied individually and in all combinations in a 2-year field experiment.

Comments:

The study was well planned and performed and it collects a series of measures on the different parameters. The topic of the research is coherent to the aims of the PLANTS journal. The work was provided with a sufficient level of scientific novelty. The authors present the study of high practical relevance.

The abstract of the paper is factual concrete, realistic, understandable, and can serve as a stand-alone document, which succinctly describes both potential use and conclusions. In my opinion, the abstract contents too many experimental details.

The introduction is informative, precise, and comprised of relevant content. The literary structure of the introduction is also good. The Paper is innovative and a new hypothesis should be formulated in the last part of the Introduction section. Include the introduction of the missing information (research gaps, importance for specific groups of plant species). Why it is required to run such specific research and in which regions? What are the alternative solutions? Se activates the antioxidant metabolism in plants to combat ROS production. What is the effect on the photochemistry of photosynthesis and photosynthetic performance?

This seems to be well-conducted research and to have clear, repeatable results.   I wish to conduct experiments in a controlled environment with a simulation of different environmental interactions.

I have no critical comments. Paper is very innovative. The structure of the paper is logical and the results are well reproduced.

Zinc (Zn) deficiency hinders the growth and development of plants and plays an important role in the regulation of photosynthesis.  In many crops, Zn deficiency showed a substantial reduction in plant biomass, photosynthetic efficiency, and cellular damage. Zn deficiency can decrease the maximum yield of PSII, photosynthesis performance index and dissipation energy per active reaction center, although the antenna size, trapping energy efficiency, and electron transport flux were stable in Zn-starved leaves. So, it is very important to investigate the impact of Zn fertilization on yield improvement and biofortification of crop plants.

The discussion lacks a bit in-depth. In the discussion, the cited information should be better compared with analogical researches. I would have expected a more critical discussion of the results.  In my opinion, authors should discuss more the regulation of the biochemical and physiological processes. The authors should add new pieces of information about the alleviation effect of selenium under drought and salinity stress. Some arguments need a tighter presentation.

The article adheres to appropriate reporting guidelines and community standards for data availability and is presented in an intelligible fashion and is written in standard English.

Conclusions are presented in an appropriate fashion and are supported by the references, however, the authors could add some new conclusions concerning future research. I also suggest using more papers from the last 5 years. That is,  the body of the text can be improved in the discussion.

I suggest adding also new references to support the interpretations. The authors are advised to read/cite the following papers related to their topics:

  • Selenium Biofortification: Roles, Mechanisms, Responses and Prospects. Molecules 2021, 26, 881. https://doi.org/10.3390/mole‐ cules26040881
  • Selenium Alleviates the Adverse Effect of Drought in Oilseed Crops Camelina (Camelina sativa L.) and Canola (Brassica napus L.). Molecules 2021, 26, 1699. https://doi.org/10.3390/ molecules26061699
  • Selenium Modulates Dynamics of Antioxidative Defence Expression, Photosynthetic Attributes and Secondary Metabolites to Mitigate Chromium Toxicity in Brassica juncea L. Plants. Environ. Expt. Bot. 2019, 161:180-192
  • Exogenous application of selenium mitigates cadmium toxicity in Brassica juncea L. (Czern & Cross) by up-regulating antioxidative system and secondary metabolites. Journal of Plant Growth Regulation 2016, 35(4): 936-950
  • Biofertilizer-Based Zinc Application Enhances Maize Growth, Gas Exchange Attributes, and Yield in Zinc-Deficient Soil. Agriculture 2021, 11, 310. https://doi.org/10.3390/ agriculture11040310
  • Zinc biofortification in rice: leveraging agriculture to moderate hidden hunger in developing countries. Archives of Agronomy and Soil Science, 2018, 64(2), pp.147-161
  • Chitosan–Selenium Nanoparticle (Cs–Se NP) Foliar Spray Alleviates Salt Stress in Bitter Melon. Nanomaterials 2021, 11, 684. https:// doi.org/10.3390/nano11030684
  • Selenium uptake, dynamic changes in selenium content and its influence on photosynthesis and chlorophyll fluorescence in rice (Oryza sativa), Environmental and Experimental Botany, 2014, vol.107, pp.39-45,
  • Zinc-biofortified wheat required only a medium rate of soil zinc application to attain the targets of 598 zinc biofortification. Archives of Agronomy and Soil Science, 2020, 1-12
  • Biofortification of wheat with zinc through zinc fertilization in seven countries. Plant and soil, 2012, 361(1), 699 119-130.
  • Cost of agronomic biofortification of wheat with zinc in China. Agronomy for Sustainable Development, 2016, 36(3), 1-7

Conclusions are too general, include and re-highlight your outcomes (research findings). Overall, the study is of good quality and the results are interesting. The manuscript can be useful for future applied research. I recommend accepting the manuscript after addressing my comments within the major revision.

Author Response

Our comments are in the attached document

Round 2

Reviewer 4 Report

The authors revised the manuscript according to the comments thoroughly and respond to the comments point by point, at present, the manuscript could be accepted.